# Human Cancer-Associated Mutations of SF3B1 Lead to a Splicing Modification of Its Own RNA

**DOI:** 10.3390/cancers12030652

**Published:** 2020-03-11

**Authors:** Tiffany Bergot, Eric Lippert, Nathalie Douet-Guilbert, Séverine Commet, Laurent Corcos, Delphine G. Bernard

**Affiliations:** 1Univ Brest, Inserm, EFS, UMR1078, GGB, F-29200 Brest, France; 2CHRU Brest, Centre de Ressources Biologiques, F-29200 Brest, France; 3CHRU Brest, Hématologie Biologique, F-29200 Brest, France; 4CHRU Brest, Génétique, F-29200 Brest, France

**Keywords:** SF3B1, splicing alterations, myelodysplastic syndromes, splice switching oligonucleotides, *Schizosaccharomyces pombe*

## Abstract

Deregulation of pre-mRNA splicing is observed in many cancers and hematological malignancies. Genes encoding splicing factors are frequently mutated in myelodysplastic syndromes, in which *SF3B1* mutations are the most frequent. SF3B1 is an essential component of the U2 small nuclear ribonucleoprotein particle that interacts with branch point sequences close to the 3’ splice site during pre-mRNA splicing. *SF3B1* mutations mostly lead to substitutions at restricted sites in the highly conserved HEAT domain, causing a modification of its function. We found that SF3B1 was aberrantly spliced in various neoplasms carrying an *SF3B1* mutation, by exploring publicly available RNA sequencing raw data. We aimed to characterize this novel SF3B1 transcript, which is expected to encode a protein with an insertion of eight amino acids in the H3 repeat of the HEAT domain. We investigated the splicing proficiency of this SF3B1 protein isoform, in association with the most frequent mutation (K700E), through functional complementation assays in two myeloid cell lines stably expressing distinct SF3B1 variants. The yeast *Schizosaccharomyces pombe* was also used as an alternative model. Insertion of these eight amino acids in wild-type or mutant SF3B1 (K700E) abolished SF3B1 essential function, highlighting the crucial role of the H3 repeat in the splicing function of SF3B1.

## 1. Introduction

About 95% of coding genes in humans are subjected to alternative splicing, a highly regulated and complex mechanism that diversifies the proteome by creating multiple proteins from the same gene. Deregulation of splicing is observed in many cancers and hematological diseases [1,2]. Recent massive sequencing of many cancer genomes allowed the identification of recurrent mutations in genes encoding splicing factors, including *SF3B1* (Splicing Factor 3B subunit 1), SRFS2 (Serine and arginine rich splicing factor 2), U2AF1 (U2 small nuclear RNA auxiliary factor 1), and ZRSR2 (Zinc finger CCCH-type, RNA binding motif and Serine/Arginine Rich 2), suggesting that somatic alterations of genes involved in splicing are common in cancer [2,3]. Strikingly, in myelodysplastic syndromes (MDS), over 50% of patients have acquired mutations that affect the splicing machinery [4,5]. MDS are complex and heterogeneous acquired pathologies of the bone marrow, characterized by a clonal and ineffective hematopoiesis, resulting in a deficit of mature myeloid blood cells and a risk of clonal progression including evolution to acute myeloid leukemia. Mutations in genes involved in epigenetic regulation are also frequently present in MDS. While many mutations may coexist in MDS and other neoplasms, splice factor mutations occur in a mutually exclusive manner [4,6] Remarkably, mutations in *SF3B1* occur in up to 85% of sideroblastic MDS (MDS-RS) [4,5,7], a subtype of MDS characterized by the presence of ring sideroblasts (RS) in the bone marrow, and associated with good prognosis. *SF3B1* mutations are also found in chronic lymphoid leukemia (CLL) [8,9], where they are associated with poor outcome, in uveal melanoma (20%) [10] and to a lesser extent in breast [11] and pancreatic cancers [12]. 

SF3B1 is an essential component of the U2 small nuclear ribonucleoprotein particle (snRNP) that interacts with branch point sequences close to the 3’ splice site during pre-mRNA splicing [13]. SF3B1 is the largest subunit of SF3b, a heptameric protein complex of U2 snRNP. SF3B1 consists of an unstructured N-terminal domain, and a C-terminal HEAT (Huntingtin, Elongation factor 3, protein phosphatase 2A, and the yeast kinase TOR1) domain composed of 20 tandem repeats structured as a superhelix [14], providing a scaffold together with other SF3b subunits within the U2 snRNP. Most of *SF3B1* mutations are heterozygous change-of-function mutations leading to amino acid substitutions at restricted sites in H4-H8 repeats of the highly conserved HEAT domain. While most of these key residues contribute to SF3B1 structure, others seem rather to be directly involved in the interactions with the intron at the branch point site and/or with spliceosomal proteins, including the K700 residue [14]. Recent RNA-sequencing (RNA-seq) analyses in various malignancies, including CLL [15] and breast cancer [16], uveal melanoma [16,17,18] and MDS [19,20,21], indicate that the most common mutations of *SF3B1* lead to a global splicing defect characterized by the production of aberrant transcripts through the use of cryptic 3’ splice sites and alternative branch points. However, the number of altered transcripts remains limited (approximately 1% of the transcriptome) likely because only introns that possess peculiar sequences appear to be sensitive to the altered form of SF3B1 [16]. Approximately half of aberrant transcripts are predicted to be degraded by the non-sense mediated mRNA decay (NMD), while the other half would be translated into proteins with potentially altered function [16]. Importantly, the differentially expressed genes and aberrant splicing events observed in cells with disease-related *SF3B1* point mutations appear to be different from those observed upon depletion of *SF3B1* [22,23], stressing the fact that SF3B1 mutants bear specific functions. Interestingly, Zhang et al. recently reported that disease-causing mutations in *SF3B1* alter splicing by disrupting interaction with the spliceosomal protein SUGP1 [24], providing a possible mechanistic explanation for the phenotype of *SF3B1* point mutants. Moreover, silencing of *SF3B1* by shRNA alters the proliferation of erythroid progenitors in vitro [23] and in vivo in a mouse model [25], highlighting its essential role during erythropoiesis. Notably, a dynamic intron retention program enriched in RNA processing genes, including *SF3B1*, regulates gene expression during terminal erythropoiesis [26,27]. However, studies on SF3B1 splice variants remain scarce. 

Upon analyzing published RNA-seq raw data, we found that *SF3B1* was aberrantly spliced in various neoplasms carrying an *SF3B1* mutation, including in MDS-RS. This aberrant transcript is expected to encode a novel SF3B1 protein isoform harboring an insertion of eight amino acids in the H3 repeat of the HEAT domain. As the functional consequences of disease-causing *SF3B1* mutations are only partially understood, we aimed to investigate whether this aberrant SF3B1 splice variant, which is a minor form, would interfere with the overall function of the spliceosome. Here, we show that cancer-associated *SF3B1* point mutations drive the formation of this SF3B1 aberrant transcript in MDS-RS patients and in human myeloid cell lines. Using complementary approaches in both human and yeast cells, we show that insertion of these eight amino acids in wild-type or mutant SF3B1 protein abolished SF3B1 essential splicing activity, highlighting the crucial role of the H3 repeat in the splicing function of SF3B1. Finally, splice-switching oligonucleotides were used to favor the formation of this peculiar SF3B1 transcript, which might represent a novel way to sensitize *SF3B1* mutated cells to splicing modulators.

## 2. Results and Discussion

### 2.1. Expression of the Main SF3B1 Mutations in MDS Patients and in Myeloid Cell Lines Drives the Formation of SF3B1ins Transcript

By exploring publicly available RNA-seq raw data obtained from patients with *SF3B1* mutations, we identified a transcript of *SF3B1* that was solely detected in *SF3B1* mutated samples in various neoplasms, including breast cancer [16], uveal melanoma [19] and recently in MDS [28]. This transcript, which we named SF3B1ins, results from the retention of 24 nucleotides from intron 12 into exon 13, through the recognition of a cryptic AG’ 3’ splice site located 24 nucleotides upstream of the canonical AG 3’ splice site (Figure 1A). As no stop codon is generated, SF3B1ins transcript is predicted to produce a protein with an insertion of eight amino acids (LLLFSLFQ) in the H3 repeat of the highly conserved HEAT domain of SF3B1 (Figure 1B). We first examined whether SF3B1ins transcript was effectively detected in a series of bone marrow mononuclear cell samples derived from 11 MDS-RS patients, in comparison to patients with Idiopathic Cytopenia of Undetermined Significance (*n* = 4) or from MDS patients without RS (*n* = 6). Most patients had normal karyotypes (Appendix A). The mutational status of a panel of 26 genes known to be frequently mutated in myeloid malignancies, including *SF3B1*, was determined by targeted NGS for MDS-RS patients (Appendix A). *SF3B1* was mutated in 9 out of 11 MDS-RS patients, with a variant allele fraction ranging from 12.3% to 48%. The two SF3B1^wt^ MDS-RS patients had mutations in the other splicing factor encoding genes *ZRSR2* or *SRSF2*, and in *TET2*. To validate splicing abnormalities in the series of MDS patients, we first analyzed the splicing of TMEM14C and ENOSF1, two genes known to present an alternative cryptic AG’ splice site efficiently selected in *SF3B1* mutated background [19,20,29]. As expected, aberrant transcripts of TMEM14C and ENOSF1 were exclusively detected in *SF3B1*-mutated MDS-RS patients. Using primers designed to specifically amplify the SF3B1ins transcript, we detected SF3B1ins systematically and exclusively in *SF3B1*-mutated samples (Figure 1C). The relative abundance of SF3B1ins was less than 10% of total SF3B1 transcripts in mononuclear cells, a proportion similar to that reported in published RNA-seq data [19]. However, the proportion of SF3B1ins protein in *SF3B1* mutated cells could not be established, as SF3B1ins cannot be distinguished from SF3B1 by Western blot. 

To investigate whether production of SF3B1ins transcript was indeed driven by *SF3B1* hot spot mutations, we first generated human myeloid leukemic cell lines (K562 and UT-7) expressing SF3B1^WT^_FLAG_ or SF3B1^K700E^_FLAG_ under the control of a doxycycline inducible promoter. We chose to study the K700E substitution, which is the most common in MDS. An optimized version of *SF3B1* was used because it was impossible to clone the natural sequence, as already reported [19]. Importantly, the amount of recombinant SF3B1_FLAG_ protein was similar to that of endogenous SF3B1 (Appendix A), mimicking a heterozygous *SF3B1*^wt/K700E^ background. We first analyzed the splicing of specific genes in different independent conditions. Aberrant splice variants of ENOSF1, TMEM14C, and DPH5 were exclusively found when SF3B1^K700E^_FLAG_ was expressed, as shown in three K562 independent clones (Figure 1D) and two independent UT-7 cell populations (Figure 1E), validating the cell models used in this study. While only traces of SF3B1ins transcript were detected in SF3B1^WT^_FLAG_-transfected cells or in the absence of transgene induction, this alternative transcript was clearly produced when SF3B1^K700E^_FLAG_ was expressed. Expression of SF3B1^K700E^ was thus sufficient to promote the formation of SF3B1ins transcript, even in the presence of endogenous SF3B1^WT^, as observed in patient cells. To check whether the SF3B1ins transcript was also produced when other *SF3B1* hot spot mutations were expressed, we transfected K562 cells with plasmids expressing SF3B1^WT^, SF3B1^K700E^, SF3B1^E622D^, SF3B1^H662Q^, or SF3B1^K666E^. Here again, SF3B1ins transcript was exclusively detected upon expression of the mutated versions of *SF3B1* (Figure 1F). Consequently, expression of the main disease-related *SF3B1* mutations is sufficient to generate SF3B1ins transcripts, even in the presence of a functional SF3B1^wt^ protein.

### 2.2. SF3B1ins and SF3B1^K700E^ins Proteins Are Defective for Splicing

To study the functionality of the SF3B1ins isoforms, we inserted the sequence encoding the LLLFSLFQ peptide into the plasmids expressing wild type SF3B1_FLAG_ or SF3B1^K700E^_FLAG_. The resulting constructs were first introduced into K562 cells by transient transfection. SF3B1^wt^_INS_ and SF3B1^K700E^_INS_ proteins were both correctly produced, although in somewhat lower abundance than SF3B1^wt^ and SF3B1^K700E^ proteins (Figure 2A), and were both properly addressed to the nucleus, as shown by immunofluorescence analysis (Appendix A). Remarkably, transient expression of SF3B1^K700E^_INS_ did not lead to the production of aberrant ENOSF1 or TMEM14C transcripts, in contrast to transient expression of SF3B1^K700E^ (Figure 2B). Thus, inserting the LLLFSLFQ sequence into the H3 repeat impeded the phenotypic expression of the K700E substitution. This may have occurred either by a mechanism whereby the LLLFSLFQ insertion in *cis* would suppress SF3B1^K700E^-dependent splicing anomalies (potentially through conformational modifications), or by inactivation of SF3B1 protein function, leading to a loss-of-function phenotype. In order to distinguish between these two hypotheses, we needed a model in which most of SF3B1 function would be assumed by the transfected constructs. This necessitated reducing the production of endogenous SF3B1, which was done by using two siRNAs directed against the 3’UTR of SF3B1. While the endogenous SF3B1 protein was efficiently depleted (Figure 2C), the FLAG-tagged SF3B1 protein was depleted in a similar way, for unknown reasons, even though there was no sequence overlap between the SF3B1 3’ UTR and the engineered SF3B1 construct. Despite this general SF3B1 decreased expression, we carried out RT-PCR experiments to study splicing events specifically altered upon downregulation of SF3B1, in order to evaluate the capacity of SF3B1^wt^_INS_ and SF3B1^K700E^_INS_ to compensate for the splicing defects caused by SF3B1 depletion. We decided to study specific exon skipping events in RBM5, DUSP11, CCNA2, and STK6, the splicing of which was known to be modified upon either SF3B1 silencing [22,30] or treatment by the spliceosome inhibitor spliceotastin A [31]. Depletion of SF3B1 in siRNA-only transfected cells or in K562 cells co-transfected with the empty vector favored skipping of exon 6 in RBM5, of exon 6 in DUSP11, of exon 5 in CCNA2 and of exons 4-5-6 in STK6 (Figure 2D,E), as previously described. While expression of both SF3B1^WT^ and SF3B1^K700E^ partially prevented exon skipping, expression of both SF3B1^wt^ins and SF3B1^K700E^ins did not. The inability of SF3B1^wt^ins and SF3B1^K700E^ins to restore the normal splicing of specific transcripts known to be altered upon silencing of SF3B1 implies that both SF3B1^wt^ins and SF3B1^K700E^ins are inactive for splicing.

To further address this question, we generated K562 cell lines that stably expressed SF3B1^WT^ins or SF3B1^K700E^ins in combination with shSF3B1, both under the control of a doxycycline-inducible promoter. Here again, the splicing pattern of RBM5, DUSP11, CCNA2, and STK6 was not rescued by expression of SF3B1^wt^ins or SF3B1^K700E^ins (Appendix A). Silencing of SF3B1 by shSF3B1 affected cell growth, as previously reported in other cell lines [32] or in CD34+ cells [23]. Notably, induction of SF3B1^wt^_INS_ or SF3B1^K700E^_INS_ did not complement the growth phenotype due to SF3B1 silencing (Figure 2F). Taken together, our results indicate that insertion of the LLLFSLFQ sequence into the H3 repeat of SF3B1 abolishes the splicing and essential function of SF3B1, underlining the crucial role of the H3 repeat in the splicing function of SF3B1. Moreover, the fact that splicing of RBM5 and DUSP11 was not affected when SF3B1^wt^ins or SF3B1^K700E^ins was expressed in regular conditions (i.e., without siSF3B1) suggests that SF3B1ins proteins do not seem to exert a dominant negative effect (Figure 2B). The effect of SF3B1ins on the pathophysiology of MDS and other neoplasms may be limited due to its low abundance. Nevertheless, while cancer-associated SF3B1 substitutions lead to an improper recognition of the branch site and to the use of cryptic splice sites, it remains possible that a fraction of the SF3B1 pool, i.e., SF3B1^wt^ins and SF3B1^K700E^ins, may drive improper U2 snRNP assembly in SF3B1^+/K700E^ cells.

Interestingly, elegant structural studies have indicated that HEAT repeat H3, as well as H5-H6 repeats, contact the β-propeller A domain of SF3B130, another core component of SF3b complex [14]. The H3 repeat seems also to be directly implicated in the interaction of the concave surface of SF3B1 with SF3b14b, a highly conserved protein required for proper assembly of yeast U2 snRNP and stability of the spliceosome [33]. SF3b14b appears to shape the structure of SF3B1 through direct contact with H2-H3 at the N-terminus of the superhelix and H15, H17, and H18 at the C-terminus of the superhelix of SF3B1. These structural features suggest that insertion of the LLLFSLFQ loop into the H3 repeat might modify the overall structure of the SF3b complex through modification of protein–protein interactions.

### 2.3. Study of Splicing Defects due to Expression of SF3B1ins or Disease-Related SF3B1 Mutations in Schizosaccharomyces pombe

To further address the question of the functionality of SF3B1ins, we performed functional complementation experiments in *Schizosaccharomyces pombe*, an alternative model of choice for the study of RNA splicing [34]. The splicing machinery is relatively well conserved between *S. pombe* and *H. sapiens*, including the presence of orthologues of SR proteins, a class of splice regulatory proteins. The *S. pombe* SF3B1 orthologue, SpSAP155 [35], which is essential for viability, shares 52% sequence identity (71% sequence identity within the HEAT domain), and 67% sequence similarity with HsSF3B1. Most substitutions found in MDS patients affect amino acids conserved between the two species, in the HEAT domain (Appendix A), including the K700 residue.

We took advantage of the thermosensitive growth phenotype of the yeast prp10-1 mutant strain [35], which harbors two point mutations in SAP155, initially selected in a genetic screen, to study the ability of various Sap155 alleles to restore growth at restrictive temperature (37 °C). We first introduced the counterparts of E622N, H662Q, K666N (R666N), and K700E substitutions into the *S. pombe* SAP155 sequence by site-directed mutagenesis. The growth deficiency of the prp10-1 strain at 37 °C was restored upon expression of wild-type SAP155 or the different mutated versions of SAP155 (Appendix A), indicating that the main counterparts of Sap155 pathogenic mutations do not affect the growth of *S. pombe*, as also observed in *S. cerevisiae* [36]. Thus, the various cancer-related SAP155 point mutations do not alter splicing of genes essential for yeast viability, in both *S. pombe* and *S. cerevisiae*. We next investigated the effect of these mutations on the ability to splice specific pre-mRNAs following a switch to the restrictive 37 °C temperature for two hours. This led to the defective splicing of TFIID, NDA3, and RPL7 in the prp10-1 strain, due to the loss of function of the endogenous sap155 (prp10-1) allele (Figure 4), as previously reported [35]. Expression of SAP155^wt^, SAP155^K700E^, or the selected main variants in the prp10-1 strain restored the correct splicing of TFIID, NDA3, and RPL7 (Figure 3 and Appendix A). Hence, yeast counterparts of disease-related SF3B1 mutants do not alter the splicing of these specific transcripts in *S. pombe*. Nevertheless, given that disease-related mutations appear to alter the fidelity of branch site selection of a mini-gene construct in *S. cerevisiae* [36], it is likely that the K700E counterpart impacts other transcripts in *S. pombe* upon selection of cryptic splice sites. The latter could be predicted by in silico analyses or identified by RNA-seq analysis, as previously done in mice. While the nature of splicing defects observed in a conditional knock-in Sf3b1^K700E/+^ mouse model mimicked that described in human MDS, with numerous aberrant 3’ splice-site selections, only a few mis-spliced transcripts were found in common between mouse and human. This resulted from the relatively poor interspecies conservation of intronic sequences able to function as aberrant splice sites [37]. Similar discrepancies are expected to be observed in *S. pombe* due to even higher divergency between yeast and human intronic sequences. 

As the insertion site of the LLLFSLFQ loop maps to an extremely conserved region of SF3B1 in the H3 repeat [14], we next introduced this loop at the exact same position in the wild-type or mutated SpSAP155 sequence to investigate how this could interfere with the function of SpSAP155 in *S. pombe*. Wild-type and prp10-1 mutant strains were transformed with plasmids expressing SAP155^WT^, SAP155^K700E^, SAP155 ^WT^ins, or SAP155 ^K700E^ins. Growth of prp10-1 cells expressing SAP155 ^WT^ins or SAP155 ^K700E^ins was affected at 25 °C. The growth phenotype was even stronger when K700E was associated to the eight-amino-acid insertion. Moreover, expression of SAP155^WT^ins and SAP155^K700E^ins in prp10-1 mutant strain did not complement at all the defective growth of prp10-1 strain at restrictive temperature (Figure 3A), in contrast to insertion-less constructions. The splicing of TFIID, NDA3, and RPL7 was not restored upon expression of SAP155^WT^ins or in prp10-1 strain (Figure 3B). Due to the major growth defect of the prp10-1 mutant strain expressing SAP155^K700E^ins, we could not study the splicing in such a background. Altogether, these results indicate that insertion of LLLFSLFQ in the H3 repeat of *S. pombe* SAP155 alters its essential function, as observed in human cell lines.

### 2.4. Use of Splice-Switching Oligonucleotides to Modulate the Production of SF3B1ins Transcript

Modulation of splicing catalysis by small molecules was proposed for therapeutic targeting of cancer cells with mutations in genes encoding spliceosomal proteins [38]. Importantly, several splicing inhibitors identified in screens for cell growth inhibitors target SF3B1, including pladienolide B, spliceostatin A, and herboxidien [39]. The pladienolide B E7107 derivative was tested in a phase 1 clinical trial in solid tumors [40] but tests were suspended due to adverse events. Recently, another pladienolide B derivative H3B-8800, which modulates both wild-type and mutant spliceosome activity, was shown to preferentially kill *SF3B1*-mutant cells. Thus, cells with altered splicing appear to be more sensitive to splicing inhibitors than normal cells, mainly by further alteration of genes involved in splicing regulation [41,42]. We speculated that increasing the splicing defects in *SF3B1*^K700E/+^ cells may make them more sensitive to splicing modulators. To test whether it was actually possible to increase SF3B1ins formation, thereby increasing splicing dysfunction in target cells, we designed splice switching oligonucleotides (SSO) with the intention to favor use of the AG’ cryptic site at the intron 12/exon 13 junction. Four different Locked Nucleic Acid SSO were chosen to mask the canonical AG splice site in different ways at this specific junction (Figure 4A). We introduced the SSO in readily transfectable HEK293T cells that were co-transfected with plasmids expressing either SF3B1^WT^ or SF3B1^K700E^, in comparison to cells transfected with a control fluorescent SSO. SSO #1, the center of which masks the 3’ AG canonical site, was able to increase SF3B1ins formation in cells expressing *SF3B1*^K700E^, but also in *SF3B1*^WT^ cells (Figure 4B,C). Nevertheless, the level of SF3B1ins in SF3B1^WT^ upon SSO #1 treatment was still lower than that observed in untreated *SF3B1*^K700E^ cells. We thus provide the proof of concept that splicing can be specifically oriented to favor the use of a cryptic AG’ splice site. Further studies are required to determine if favoring such an event could increase the sensitivity to splicing modulators, including the promising H3B-8800 compound. A similar approach that would specifically orient the spliceosome to a specific cryptic splice site in other splicing factor encoding genes might be exploited in the future to make *SF3B1*^K700E/+^ cells more sensitive to splicing modulators.

## 3. Materials and Methods

### 3.1. Patient Samples

Bone marrow samples were obtained from patients followed up in Brest University Hospital and in St. Brieuc Hospital, and collected in the Centre de Ressources Biologiques (CRB) of Brest Hospital, which is referenced by the BioBank structure under a unique identifier number BB-0033-00037. The CRB is certified according to the French norm NF S 96-900: “CRB management system and quality of biological resources.” This was a non-interventional retrospective study approved by the ethics committee of Brest Hospital (ethic code 29BRC20.0029). Mononuclear cells were isolated by Ficoll gradient from bone marrow aspirates of patients diagnosed with MDS according to the World Health Organization classification, or from individuals with normal bone marrow. Written informed consent was obtained from all subjects, and the experiments conformed to the principles set out in the Declaration of Helsinki.

### 3.2. Next-Generation Sequencing

The next-generation sequencing panel included 26 genes (*ASXL1*, *BCOR*, *CBL*, *CSF3R*, *DNMT3A*, *ETNK1*, *ETV6*, *EZH2*, *IDH1*, *IDH2*, *JAK2*, *KRAS*, *MPL*, *NRAS*, *PDGFRA*, *RUNX1*, *SETBP1*, *SF3B1*, *SH2B3*, *SRSF2*, *STAG2*, *TET2*, *TP53*, *U2AF1*, *ULK1*, and *ZRSR2*) and the sequencing was performed using Ampliseq^TM^ (Thermo Fisher Scientific, Foster City, CA, USA) custom design. Library preparation and sequencing using Ion PGM^TM^ (Thermo Fisher Scientific) were performed according to the manufacturer’s instructions. Mutations were detected using the Variant Caller v4.2 plugin from Torrent Suite Software and IonReporter v5.2 (Life Technologies, Carlsbad, CA, USA). For mutation calling, arbitrary filters were fixed with variant allele frequencies >2% and depth >50X. False positive variants were dropped after BAM analysis on Alamut (Interactive Biosoftware, Rouen, France). Only exonic non-synonymous mutations were analyzed.

### 3.3. Cloning and Site-Directed Mutagenesis

Due to the impossibility of cloning the natural sequence of *SF3B1* cDNA, a codon-optimized version of *SF3B1* (from SF3B1-201 ENST00000335508.10), originally cloned in pCMV-3tag-1A vector [19], was used for all experiments. pCMV-3tag-1A-*SF3B1*^WT^ and pCMV-3tag-1A-*SF3B1*^K700E^ were generously given by Angelos Constantinou. Plasmids encoding E622D, H662Q, and K666E variants of SF3B1 were obtained by site-directed mutagenesis from pCMV-3tag-*SF3B1*^WT^ plasmid using the QuikChange II XL Site-Directed Mutagenesis Kit (Agilent). The same strategy was used to produce the counterparts of E622N, H662Q, K666N (R666N), and K700E substitutions into the *S. pombe SAP155.* SF3B1ins coding sequence was created by pseudo-inverse PCR using specific primers to incorporate the 24 nucleotides corresponding to the retained part of the intron 12 in the optimized sequence of *SF3B1* (*SF3B1*^WT^ and *SF3B1*^K700E^), in both pCMV-3tag-1A and pCW57.1 plasmids. Primers used for cloning and directed mutagenesis are listed in Appendix A.

### 3.4. Cell Culture and Transfection

The K562 and UT-7 cell lines were obtained from the American Type Culture Collection (ATCC) and the German Collection of Microorganisms and Cell Cultures (DSMZ), respectively. K562 and K562-derived cell lines were cultured in RPMI 1640 medium (Gibco), supplemented with 10% fetal bovine serum (Gibco) and 2 mM of L-glutamine (Gibco), at 37 °C and 5% of CO_2_. UT-7 cells were cultured in IMDM medium (Gibco), 10% fetal bovine serum (Gibco), 4 mM of L-glutamine (Gibco), 100 UI/mL of penicillin and streptomycin (Gibco), and 1 U/mL of erythropoietin (EPO). K562 cells were transfected by electroporation using Cell Line Nucleofector^TM^ Kit V (Amaxa, Lonza) according to the manufacturer’s instructions. Transfections were done with 2 × 10^6^ cells and 4 μg of plasmid and/or 30 pmol of *SF3B1* siRNAs designed to specifically target the 3’UTR sequence (si*SF3B1*-1: 5’-GUGUAGAACUGGUCAUAGA-3’, si*SF3B1*-2: 5’-CUCAUUCCUUGUGUUUAAA-3’). HEK293T cells were cultured in EMEM medium (Gibco) supplemented with 10% fetal bovine serum (Gibco) and were transfected with lipofectamine 3000 (Invitrogen) according to the manufacturer’s instructions. Transfections were done with 400,000 cells, 1 μg of plasmid and 500 pmol of SSO. SSO were synthetized and modified by the locked nucleic acid technology (Qiagen). SSO control was labeled with a 5’FAM to estimate the transfection efficacy. (SSO control: 5’-TAACACGTCTATACGCCCA-3’; SSO #1: 5’-CCACGAGGATCTGAAAAA-3’; SSO #2: 5’-ATGACCAGGATCTGA-3’; SSO #3**: 5’-TTCAATGACCACGAGGAT-3’; SSO #4 : 5’-CTGAAAAAGAGAAAGAG-3’). All analyses were performed at 48 h post-transfection. 

### 3.5. Generation of SF3B1 and shSF3B1 Inducible Cell Lines

The optimized coding sequences of *SF3B1*^WT^ and *SF3B1*^K700E^, amplified from pCMV-3tag-1A-*SF3B1*^WT^ and pCMV-3tag-1A-*SF3B1*^K700E^ plasmids, respectively, were introduced in the lentiviral plasmid pCW57.1 using In-Fusion® HD Cloning Plus kit (Takara). sh*SF3B1* targeting endogenous SF3B1 mRNA was cloned into the pLKO-Tet-on plasmid, an all-in-one inducible system. Sequences of the oligonucleotides used for sh*SF3B1* (#5) were as described in [32], and did not overlap with the optimized coding sequence of *SF3B1*. 

Forward: CCGGCCTCGATTCTACAGGTTATTACTCGAGTAATAACCTGTAGAATCGAGGTTTTTG, 

Reverse: AATTCAAAAACCTCGATTCTACAGGTTATTACTCGAGTAATAACCTGTAGAATCGAGG 

Lentival particles were produced by the Vect’UB vectorology core facility (Bordeaux, France). Cells were transduced by lentivirus particles with a multiplicity of infection of 5. Selection of stable positive cells was done with puromycine (Sigma) for *SF3B1* inducible cell lines or G418 (Sigma) for sh*SF3B1* inducible cell lines. Transduction with the sh*SF3B1* construct was done on both K562 and K562 stable cell lines expressing SF3B1^wt^, SF3B1^K700E^, SF3B1^wt^ins, and SF3B1^K700E^ins to obtain co-transduced stable cell lines. *SF3B1* expression was induced by treating cells with 2 μg/mL or 3 μg/mL (sh*SF3B1*) of doxycycline (Sigma). All analyses were performed 48 h after induction with doxycycline. To determine cell growth, cells were counted every 24 h following doxycycline induction, using a hemocytometer.

### 3.6. RNA Extraction and RT-PCR

RNA was extracted from bone marrow mononuclear cells of MDS patients using the Nucleospin RNA kit (Macherey-Nagel) or TRIzol (Invitrogen)–phenol/chloroform extraction. RNA was extracted from K562 and HEK293T cells using nucleospin RNA kit (Macherey-Nagel) or nucleospin RNA XS (Macherey-Nagel). RNA integrity was checked using Bioanalyser or by agarose gel electrophoresis. Reverse transcription was performed with High Capacity cDNA reverse transcription kit (Applied Biosystems). PCR was performed with the GoTaq® G2 DNA Polymerase kit (Promega). Primers used are listed in Appendix A. PCR products were analyzed on 2% or 3% agarose gels (Molecular Biology Grade Agarose, Eurobio). Quantification of DNA bands was performed with Image studio lite software. Sequencing of extracted gel bands was achieved to confirm the partial intron retention observed in SF3B1ins transcript. Quantitative real-time PCR was performed with Power SYBR™ Green PCR Master Mix with a StepOnePlus Real-Time PCR System. The 2^∆∆-Ct^ method was used to analyze the results using *GAPDH* as a reference gene.

### 3.7. Western Blot Analysis

Proteins were extracted from cells with a lysis buffer (NP40 1%, SDS 0.1%) and quantified with the Dc protein kit (Biorad). Forty μg of total proteins were resolved by SDS-polyacrylamide gel electrophoresis (SDS-PAGE) and transferred onto nitrocellulose membranes. We used the following primary antibodies: anti-SF3B1 (1:5000, #170854 Abcam), anti-FLAG (1:5000, #F3165, Sigma), anti-GAPDH (1:5000), and anti-b-actin (1:1000, #ab8226, Abcam). Donkey anti-mouse (1:5000, #926-32222, LI-COR) and goat anti-rabbit (1:5000, #926-32211, LI-COR) secondary antibodies were used. Immunofluorescence was detected using Odyssey Infrared Imaging system (LI-COR). Quantification was performed with Image Studio Lite software. β-actin or GAPDH were used as references to normalize quantification.

### 3.8. Yeast Growth and Molecular Biology

Wild-type and *prp10-1 Schizosaccharomyces pombe* strains [35] were transformed with pREP41-HA plasmids using a slightly modified version of the lithium acetate method [43] in which heat shock was replaced by an hour-long incubation at 25 °C due to the thermosensitive phenotype. Transformants were selected through their ability to grow on leucine free EMM 2% glucose medium (Formedium). To analyze splicing defects, cells were cultured in EMM 2% glucose at 25 °C, and subjected to a 2-h shift at 37 °C (restrictive temperature). Cell wall was broken by vortexing cells for 20 min with acid-washed glass beads (Sigma) at 4 °C and RNA was extracted using Nucleo-spin RNA kit (Macherey-Nagel). Reverse transcription and RT-PCR were performed as described in the preceding section, using primers allowing detection of intron retention in TFIID, NDA3, RPL7 (Appendix A).

### 3.9. Statistical Analysis

Statistical analysis and histograms were created with Prism 8 software. When appropriate, a Mann–Whitney test was used with a *p*-value < 0.05 to be considered significantly different between two groups.

## 4. Conclusions

RNA splicing was reported as a process significantly affected (i.e., numerous splicing genes with abnormal expression or splicing) in all MDS harboring mutations in splice factor genes [44]. In this study, we identified and characterized a novel SF3B1 transcript that is specifically produced in neoplasms carrying an SF3B1 mutation, including in myelodysplastic syndromes. This transcript encodes an aberrant SF3B1 protein isoform with an insertion of the LLLFSLFQ sequence into the H3 repeat of the HEAT domain. Using complementary approaches in both human and yeast cells, we showed that this insertion abolished SF3B1 splice function, highlighting the functional importance of the H3 repeat in SF3B1. SF3B1ins transcript level might vary differently in a disease-specific manner, and modulate the phenotypical expression of SF3B1 point mutants. SF3B1 mutant cells are expected to accumulate numerous other aberrant proteins, which could participate in carcinogenesis. So far, to our knowledge, only two studies dealt with the neoproteins specifically produced in SF3B1 mutated cells ([28] and this work). Significant progress is expected in the characterization of these splicing-derived cancer specific proteins in order to propose novel biomarkers and therapeutic options. Finally, splicing inhibitors were recently proposed for therapeutic targeting of cancer cells with mutations in genes encoding spliceosomal proteins. As the selectivity still remains an important issue, increasing splicing alterations in SF3B1^K700E/+^ cells in a specific way, through preferential selection of splice sites by SSO, might represent a novel way to sensitize cells to splicing inhibitor-induced apoptosis. 

## Figures and Tables

**Figure 1 cancers-12-00652-f001:**
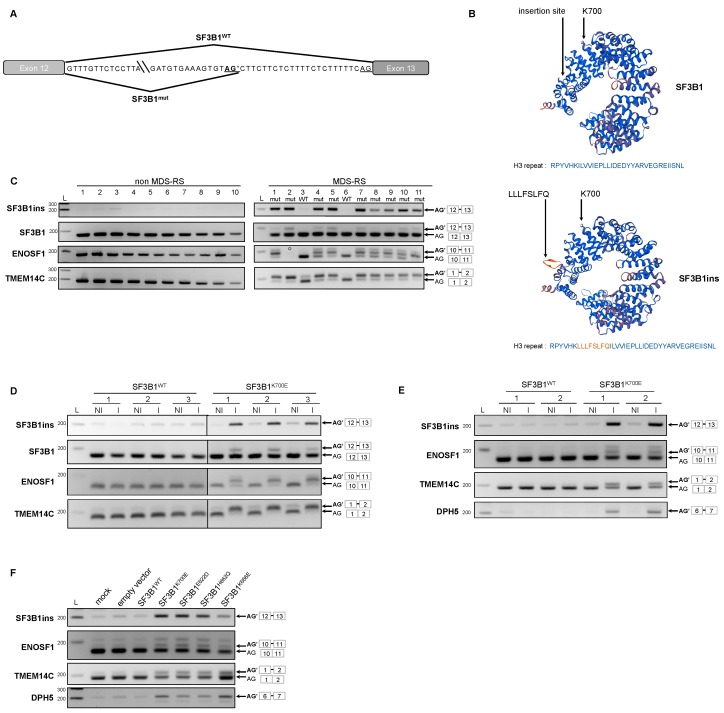
A novel splicing isoform of *SF3B1*, SF3B1ins, is detected in sideroblastic myelodysplastic syndromes (MDS-RS) patients and myeloid cell lines expressing mutant *SF3B1*. (**A**) Representation of the aberrantly spliced junction of *SF3B1*. (**B**) 3D modeling of the eight amino acid insertion in the H3 repeat of SF3B1 Huntingtin, Elongation factor 3, protein phosphatase 2A, and the yeast kinase TOR1 (HEAT) domain. The site of insertion and the lysine in position 700 are indicated in both normal (top) and predicted (bottom) structures. Primary sequences of normal and aberrant H3 repeat (amino acids 588–605) are indicated below the 3D structures. The inserted sequence is colored in orange. (**C**) Detection of SF3B1ins transcript in mononuclear cells from myelodysplastic syndromes (MDS) patients harboring *SF3B1* mutations. RNA was extracted from mononuclear cells of subjects with normal bone marrow (lanes 1–4), from MDS patients without RS (lanes 5–10) and from MDS-RS patients. The mutational status of *SF3B1* is indicated for MDS-RS patients, as follows: mut—mutated; WT—wild-type. RT-PCR was performed using primers allowing specific detection of SF3B1ins or detection of both aberrant (upper band) and canonical (lower band) transcripts of SF3B1, ENOSF1 and TMEM14C. (°: ENOSF1 RT-PCR on patient 2 could not be performed due to an insufficient quantity of material); (**D**,**E**) Inducible expression of SF3B1^K700E^ in K562 cells and UT-7 cells generates SF3B1ins transcript. RT-PCR was performed from K562 cells (**D**) and UT-7 (**E**), expressing SF3B1^WT^ (left) or SF3B1^K700E^ (right) (I: induced; NI: non induced), using primers as described in C. Steady-state SF3B1 protein level was achieved by Western blot (Appendix A); (**F**) Detection of SF3B1ins transcript in K562 transiently expressing distinct *SF3B1* variants. RT-PCR was performed using primers as described in C. Data information: In (**D**,**E**), representative results of at least three independent RT-PCR experiments are presented. L: ladder.

**Figure 2 cancers-12-00652-f002:**
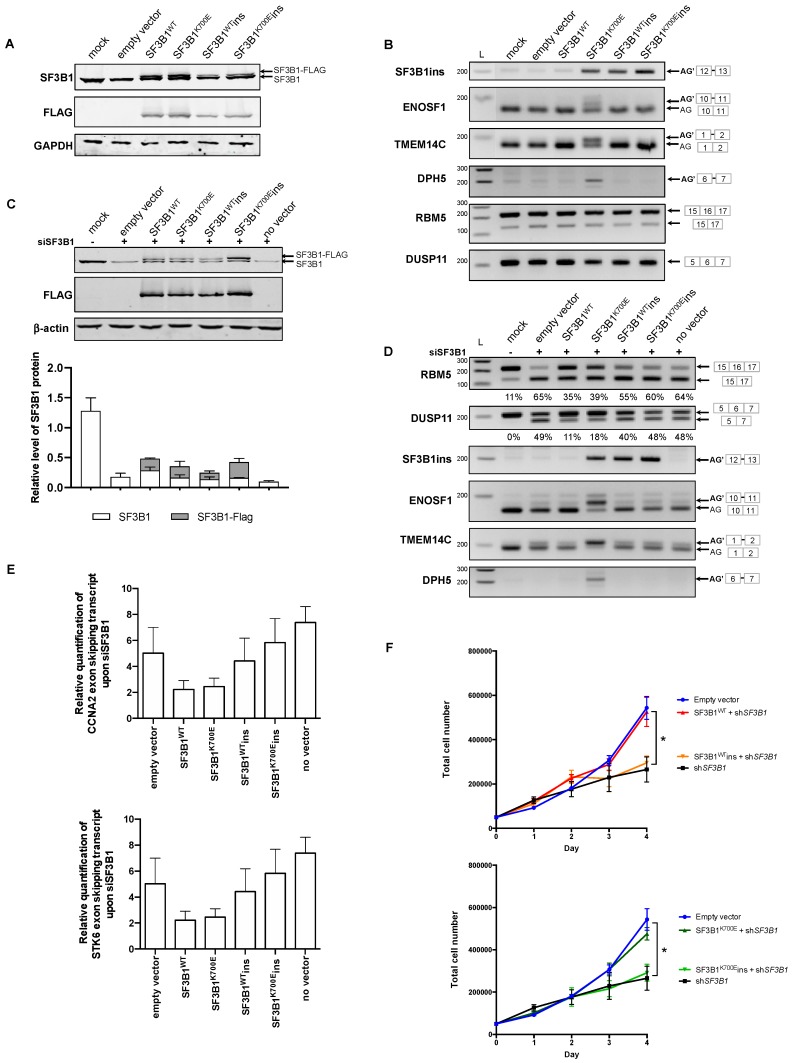
SF3B1ins protein is defective for splicing. (**A**–**E**) K562 cells were transfected with plasmids encoding different versions of SF3B1 (SF3B1^WT^, SF3B1^K700E^, SF3B1^WT^ins, and SF3B1^K700E^ins). (**A**) Total SF3B1 proteins (endogenous and exogenous) were detected by Western blot analysis using anti-SF3B1 antibody. Plasmid-encoded SF3B1 was detected using anti-FLAG antibody. (**B**) Analysis of splicing events known to be specifically altered in *SF3B1*-mutated background (ENOSF1, TMEM14C, DPH5) or upon *SF3B1* loss of function (RBM5, DUSP11). RT-PCR was performed using primers allowing specific detection of distinct splicing events in SF3B1, ENOSF1, TMEM14C, DPH5, RBM5, and DUSP11. (**C**–**E**) K562 cells were co-transfected with an siRNA specific to endogenous SF3B1 and with plasmids encoding different versions of SF3B1. (**C**) Western blot analysis of SF3B1 proteins as described in (A). Average quantification of endogenous and exogenous levels of SF3B1 protein from three independent experiments is indicated. (**D**) Analysis of specific splicing events as described in (**B**). The proportion of exon skipping in RBM5 and DUSP11 (average from three independent experiments) is indicated below the corresponding gels. (**E**) RT-qPCR was performed using primers allowing specific quantification of exon skipping events in CCNA2 (exon 5) and STK6 (exons 4, 5, and 6) transcripts, normalized to GAPDH. Relative quantification is indicated, and error bars represent ± SEM from three independent experiments. (**F**) Growth of K562 cells expressing inducible SF3B1^WT^ versus SF3B1^WT^ins (top) and SF3B1^K700E^ versus SF3B1^K700E^ins (bottom) upon *SF3B1* silencing (shSF3B1), following doxycycline induction. Error bars represent ± SEM from four independent experiments. A Mann–Whitney test was applied (*p* value < 0.05). Data information: In (**A**–**D**), representative results of at least three independent experiments are presented. In (C and E), error bars represent ± SEM from three independent experiments. L: ladder.

**Figure 3 cancers-12-00652-f003:**
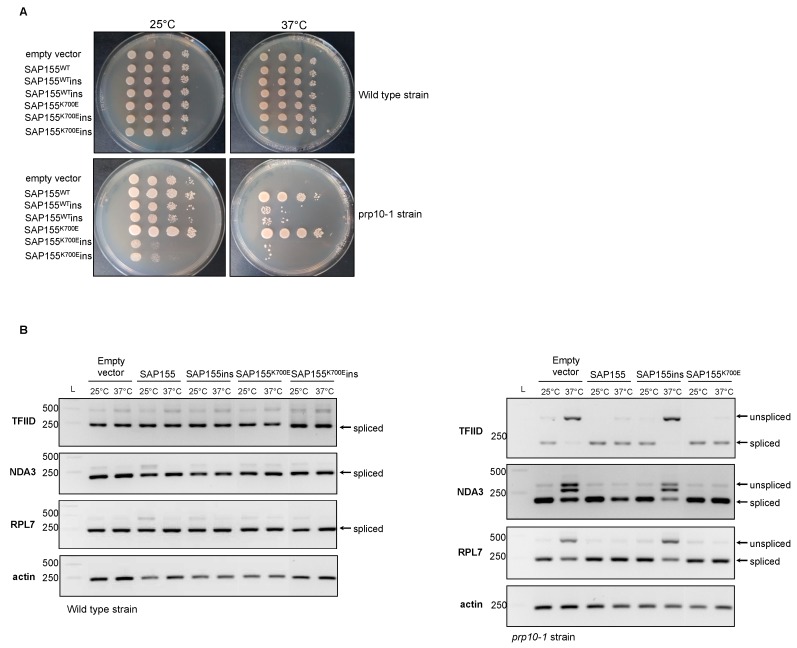
Inserting the eight amino acids in the HEAT H3 repeat of yeast *Schizosaccharomyces pombe* SF3B1 ortholog abolishes its function. (**A**) Growth of *prp10-1* mutant cells expressing wild-type or mutated versions of SpSAP155. *prp10-1* mutant cells were transformed with the following yeast expression plasmids: pREP41, pREP41-HA-SAP155, pREP41-HA-SAP155^K700E^, pREP41-HA-SAP155ins, or pREP41-HA-SAP155^K700E^ins. Ten-fold serial dilutions of yeast transformants were deposited on EMM 2% glucose media and cultured at 25 °C or 37 °C. (**B**) RT-PCR splicing analysis of TFIID, NDA3, and RPL7 transcripts in the *prp10-1* strain expressing different versions of SpSAP155. Transformants were cultured at 26 °C and subjected to a temperature switch at 37 °C for 2 h. RNA was extracted and analyzed by RT-PCR using specific primers allowing the detection of both spliced and unspliced mRNA. Data information: In (**A**,**B**), representative results of three independent experiments are presented. L: ladder.

**Figure 4 cancers-12-00652-f004:**
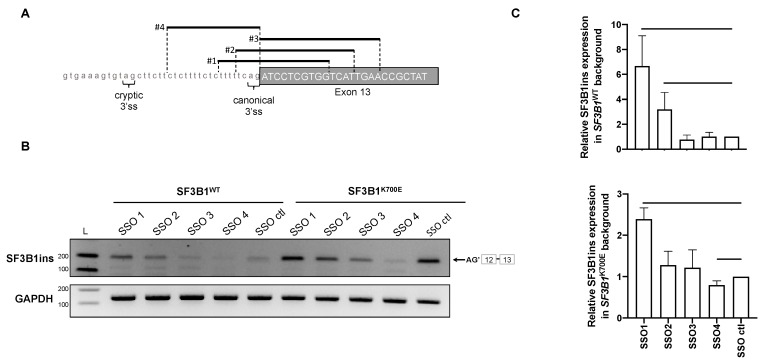
The level of SF3B1ins transcript can be modulated by splice-switching oligonucleotides. (**A**) Position of four splice-switching oligonucleotides at the intron 12–exon 13 junction of *SF3B1*. (**B**,**C**) Detection of SF3B1ins transcript in HEK293T cells transfected with plasmids encoding SF3B1^WT^ or SF3B1^K700E^ together with splice-switching oligonucleotides (SSO), by classical RT-PCR (**B**) and by RT-qPCR (**C**). The effect of SSO in either *SF3B1*^WT^ or *SF3B1*^K700E^ was measured by RT-PCR and RT-qPCR, which were performed using primers allowing specific detection of SF3B1ins, in comparison to a control (ctl) SSO. SF3B1ins quantification by RT-qPCR was normalized to GAPDH. Data information: In (**C**), error bars represent ± SEM from three independent experiments. A Mann–Whitney test was applied.

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
