# Peer review of "Human Cancer-Associated Mutations of SF3B1 Lead to a Splicing Modification of Its Own RNA"

_cancers, 2020, doi:10.3390/cancers12030652_

Round 1

Reviewer 1 Report

In the study entitled “Human cancer-associated mutations of SF3B1 lead to a splicing modification of its own RNA”, authors found mutated SF3B1 can feedback to regulate alternative splicing of SF3B1 itself in human cancer cell line. They also found the isoform SF3B1ins play role in biological means. The results are very interesting. Please see below for my comments.

  1. I’d like to know more details about mechanism of SF3B1ins in cancer. For example, is it regulating splicing events of cancer related genes?
  2. Authors claimed that “Mutations in genes involved in epigenetic regulation are also frequently present in MDS”. While I didn’t see epigenetic part in results section.
  3. I am not very understand why combining human cancer results with yeast results?
  4. Signals in Figure 2F are mixed together, it will be better if author divided them into two or more panels.
  5. Manuscript have several typos, e.g., font of gene name should be consistent in Italian.

Author Response

Point 1. I’d like to know more details about mechanism of SF3B1ins in cancer. For example, is it regulating splicing events of cancer related genes?

Here we show that the SF3B1ins isoform is defective for splicing. The proportion of SF3B1ins produced in cancer cells harboring SF3B1 mutations may vary and modulate the effect of cancer-associated mutations of SF3B1. Splicing of cancer related genes, such as Mcl-I, MDM2 or RBM5, is known to be altered upon silencing or inhibition of SF3B1 (Larrayoz et al., 2016 ; Wu et al. 2019 ; Corrionero et al. 2011 ). Nevertheless, the weak proportion of defective SF3B1 in SF3B1K700E/+ cells is most probably not sufficient to induce such aberrant splicing events.

Point 2. Authors claimed that “Mutations in genes involved in epigenetic regulation are also frequently present in MDS”. While I didn’t see epigenetic part in results section.

This sentence was added to provide a general background of the main mutations frequently found in MDS. This sentence can be removed if this is the preference of the editorial board.

Point 3. I am not very understand why combining human cancer results with yeast results?

Given that the insertion site of LLLFSLFQ sequence maps a highly conserved region of SF3B1, we used Schizosaccharomyces pombe as an alternative model to stress the deleterious effect of this insertion into the H3 repeat of HEAT domain.

Point 4. Signals in Figure 2F are mixed together, it will be better if author divided them into two or more panels.

We have divided Figure 2F into two panels to increase readability of the figure.

Point 5. Manuscript have several typos, e.g., font of gene name should be consistent in Italian.

This has been corrected in the new version of the manuscript.

Reviewer 2 Report

Bergot and colleagues reported about a RNA isoform with eight additional amino acids in the H3 repeat of the highly conserved HEAT domain in MDS patients with mutant SF3B1. Despite of the low abundance of SF3B1ins RNA in comparison with total SF3B1 transcript level (and probable in the protein level), the SF3B1ins isoform is responsible for differential alternative splicing in neoplasms.

This manuscript is well-written and structured. It provides many experimental data and models in human cell lines and yeast strains to contribute to our understanding of the pathophysiology of SF3B1 mutations.

Minor point: Can the Authors explain why traces of SF3B1ins are shown in there SF3B1wtFLAG-transfected cells? (line 128 f.)

Author Response

Minor point: Can the Authors explain why traces of SF3B1ins are shown in there SF3B1wtFLAG-transfected cells? (line 128 f.)

Traces of SF3B1ins are indeed detected upon expression of SF3B1wt or in the absence of transgene induction (as mentioned in line 129). Thus, the AG’ cryptic site of this specific junction can be selected by wild type SF3B1 albeit at a very low frequency.

Reviewer 3 Report

The manuscript “Human cancer-associated mutations of SF3B1 lead to 2 a splicing modification of its own RNA,” by Bergot et. Al., present strong experimental evidence of the aberrant splicing of SR3B1. The study utilized the mammalian and S. pombe cells to repeat their study in two experimental models. The study demonstrates that the aberrant SF3B1 splice variant potentially interferes with the spliceosome fidelity resulting in dysregulation of global pre-mRNA splicing. The study illustrates the experiments and methods in detail with appropriate and necessary controls. The narrative, as well as figures and legends, clearly and elegantly convey the manuscript's goal and purpose. The supplementary data is appropriate and supports the conclusions.

Overall the manuscript is very well written, but there are just a few minor weaknesses that need to be addressed before publication.

  • In line 61, please make it more explicit about “the most common mutations of SF3B1 lead to aberrant transcripts.” The experts would understand that you are referring global splicing but readers from peripheral interest areas would have difficulty following it.
  • Figure 1 B shows the structural model of the H3 repeat of the SF3B1 HEAT domain. While it is nice, but please add a figure illustrating the primary sequence of the region also should the insertion.
  • Figure 1 F (also in Figure 2 D) Do you have any thoughts on the faster-running band in DPH5 and slower running band in ENOSF1?
  • Please state if the sequencing of the extracted gel bands was determined from a representative lane to determine/confirm the intron retentions.
  • It is not clear which methods were used to determine cell growth. Please include a section in the materials and methods.
  • Section 4.3 should be Cloning and “Site” directed mutagenesis.
  • Expand the acronyms “ATCC and DSMZ” for reader convenience.
  • What type of Agarose (and the supplier) was used for splicing phenotype fractionation?
  • Many NGS platforms are using different sequencing technologies available. Can you be specific?

Author Response

Point 1- In line 61, please make it more explicit about “the most common mutations of SF3B1 lead to aberrant transcripts.” The experts would understand that you are referring global splicing but readers from peripheral interest areas would have difficulty following it.

To clarify this point, we introduced this modification : “the most common mutations of SF3B1 lead to a global splicing defect characterized by the production of aberrant transcripts through the use of cryptic 3’ splice sites and alternative branch points”.

Point 2- Figure 1 B shows the structural model of the H3 repeat of the SF3B1 HEAT domain. While it is nice, but please add a figure illustrating the primary sequence of the region also should the insertion.

To address this point, we have added the primary sequence of the H3 repeat – with and without the insertion - below the structural models in Figure 1B.

Point 3- Figure 1 F (also in Figure 2 D) Do you have any thoughts on the faster-running band in DPH5 and slower running band in ENOSF1?

The slower-running band in ENOSF1, which has been previously described by Alsafadi and colleagues, corresponds to the heteroduplex formation from two bands (AG and AG’)(Alsafadi et al.,  2016). The size of the faster-running band in DPH5 corresponds to that of a DPH5 alternative transcript with a shorter exon 5 (44 nucleotides shorter, referenced as ENST00000464270.5).

Point 4 - Please state if the sequencing of the extracted gel bands was determined from a representative lane to determine/confirm the intron retentions.

Sequencing of extracted gel bands was achieved to confirm the partial intron retention observed in SF3B1ins transcript. This sentence has been added in part 4.6. of the manuscript.

Sequencing of extracted gel bands has not been done for TMEM14C, ENOSF1 and DPH5 controls, as these events have been described in previous studies (Alsafadi et al., Dolatshad et al. 2016)

Point 5- It is not clear which methods were used to determine cell growth. Please include a section in the materials and methods.

We have added the following sentence in section 4.5. : « To determine cell growth, cells were counted every 24 hours following doxycycline induction, using a hemocytometer. »

Point 6 - section 4.3 should be Cloning and “Site” directed mutagenesis.

This has been done.

Point 7- Expand the acronyms “ATCC and DSMZ” for reader convenience.

We have expanded the acronyms in the « Material and Methods » section 4.4., as followed : ATCC American Type Culture Collection ; DSMZ German Collection of Microorganisms and Cell Cultures

Point 8- What type of Agarose (and the supplier) was used for splicing phenotype fractionation?

We have specified the type and the supplier of Agarose (Molecular Biology Grade Agarose, Eurobio) in section 4.6.

Point 9- Many NGS platforms are using different sequencing technologies available. Can you be specific?

We have used Ion PGM from Thermo Fisher Scientific (Ion 316TM Chip v2), which is more broadly known as Ion Torrent technology. We have added « Ion » in section 4.2.

Round 2

Reviewer 1 Report

Authors addressed my concerns adequately.